# The Creation of the Suppressive Cancer Microenvironment in Patients with HPV-Positive Cervical Cancer

**DOI:** 10.3390/diagnostics12081906

**Published:** 2022-08-06

**Authors:** Katarzyna Chaberek, Martyna Mrowiec, Magdalena Kaczmarek, Magdalena Dutsch-Wicherek

**Affiliations:** 12nd Department of Obstetrics and Gynaecology, Center of Postgraduate Medical Education (CMKP), 01-813 Warsaw, Poland; 2Department of Endoscopic Otorhinolaryngology, Center of Postgraduate Medical Education (CMKP), 01-813 Warsaw, Poland

**Keywords:** suppressive cancer microenvironment, cervical cancer, HLA-G antigen expression

## Abstract

The development of malignancy is closely connected with the process of cancer microenvironment remodeling. As a malignancy develops, it stimulates the creation of the suppressive microenvironment of the tumor through the presence of cells that express membrane proteins. These proteins are secreted into the cancer microenvironment, where they enable tumor growth. In patients with cancer of the cervix, the development of the disease is also linked to high-risk HPV (hr-HPV) infection. Such infections are common, and most clear spontaneously; however, a small percentage of these infections can persist and progress into precancerous cervical intraepithelial neoplasia and invasive cervical carcinoma. Consequently, it is assumed that the presence of hr-HPV infection alone is not sufficient for the development of cancer. However, chronic HPV infection is associated with the induction of the remodeling of the microenvironment of the epithelium. Furthermore, the local microenvironment is recognized as a cofactor that participates in the persistence of the HPV infection and disease progression. This review presents the selected immune evasion mechanisms responsible for the persistence of HPV infection, beginning with the delay in the virus replication process prior to the maturation of keratinocytes, the shift to the suppressive microenvironment by a change in keratinocyte immunomodulating properties, the alteration of the Th1/Th2 polarization of the immune response in the microenvironment, and, finally, the role of HLA-G antigen expression.

## 1. Introduction

The development of malignancy is closely connected with the process of cancer microenvironment remodeling. As a malignancy develops, it stimulates the creation of the suppressive microenvironment of the tumor through the presence of cells that express membrane proteins. These proteins are secreted into the cancer microenvironment, where they enable tumor growth [1,2,3]. In patients with cancer of the cervix, development of the disease is also related to the presence of high-risk HPV (hr-HPV) infection [4]. Although hr-HPV infections are common and approximately 85–90% clear spontaneously, 10–15% can persist and progress into precancerous cervical intraepithelial neoplasia and invasive cervical carcinoma [5]. Therefore, it is assumed that the presence of hr-HPV infection alone is not sufficient for the development of cancer [6]. HPV infections must escape both the adaptive and the innate immune responses. Their life cycle is intraepithelial, without viremia, and without inducing inflammation. While the infection persists, proinflammatory cytokines are not released and dendritic cells are not recruited and activated [7]. The development of cancer is closely related to the remodeling of the microenvironment of the epithelium under chronic HPV infection. The local microenvironment is recognized as a cofactor that participates in the persistence of the HPV infection and in disease progression. Various studies have suggested that HPV-infected cells actively remodel the local microenvironment, enabling the persistence of infection and promoting the chronic inflammation that supports cervical neoplasia [5,8]. 

### 1.1. HPV

Human papillomaviruses (HPVs) belong to the Papillomaviridae family that comprises epitheliotropic DNA viruses with an affinity for squamous epithelia. These viruses infect the skin and mucosae, inducing benign hyperproliferative lesions, such as warts and asymptomatic precursor lesions, that can progress to high-grade neoplasia and malignancy. Certain types of papillomaviruses are also known to show a preference for distinct anatomical sites [8,9,10,11]. The HPV family of viruses is divided into two main groups based on their oncogenic abilities: “low-risk HPVs” and “high-risk HPVs” [10]. The association of high-risk HPV (hr-HPV) types with the development of cervical and oral cancers has been well established. HPV-16 is the most common oncogenic type of HPV. The next most common are HPV-18, -52, -31, -58, -39, -56, and -51, respectively. This information has provided a rationale for the introduction of HPV DNA testing in cervical screening, as well as the development of prophylactic vaccines against HPV-16 and -18, the most common oncogenic types of papillomaviruse responsible for cervical cancer [4,12,13]. Most HPV infections are either asymptomatic or resolve spontaneously, with a clearance time of 6–24 months. The interval from the acquisition of carcinogenic HPV infection until a CIN2/3 (cervical intraepithelial neoplasia) lesion forms is estimated to be, on average, from 5 to 10 years. By contrast, the period for the development of cancer from HPV infection is estimated to be from 10 to 20–25 years [14,15]. HPV is responsible for about 5% of all cancer cases globally, and approximately 3–3.5% of all cases of HPV-related cancers are caused by HPV-16 [16]. HPV-16 and -18 cause approximately 70% of all cases of cervical cancer, with HPV-16 responsible for about 55–60% of these cases and HPV-18 for about 10–15%. The other hr-HPV genotypes contribute to most of the remaining 30% of cervical cancer cases while a small percentage of cases are attributable to other high-risk, and even low-risk, HPV genotypes [11,17]. Of HPV-attributable cancer cases, 80% are cervical cancer cases, with women thus bearing around 90% of HPV-attributable cancers worldwide. These proportions also vary according to income group; the highest percentages of these cases are found in the lowest-income countries. Conversely, in high-income countries, the proportion of the HPV-attributable cancer burden due to other anogenital and head and neck cancers is higher than in low-income countries, as is the proportion of HPV-attributable cancer occurring in men [18]. HPV types 16 and 18 are estimated to account for 72% of all HPV-attributable cancer cases; HPV-31, -33, -45, -52, and -58 account for an additional 17% of such cases, most of which represent cervical cancer. In men, HPV-16 and -18 are responsible for virtually all HPV-attributable cancers [18]. Moreover, there is no evidence of a significant genetic predisposition for cervical cancer [19]. Around 30% of all oropharyngeal cancers, comprising mainly those occurring at the site of the tonsils and the base of the tongue, are caused by HPV, with HPV-16 and -18 responsible for 85% of these cancer cases [13,20]. 

### 1.2. HPV’s Transforming Potential

The transforming potential of the HPV virus is determined by HPV E6 and E7 oncoproteins. These oncoproteins are responsible for the inactivation and degradation of both the tumor suppressor protein p53 (E6) and the retinonblastoma protein (Rb). E6 proteins that bind p53 stimulate its degradation [21]. The P53 gene encodes transcription factor and has the ability to induce cell-cycle arrest and apoptosis [22]. In normal cells, p53 activity is low; however, p53 levels can rise in response to DNA damage, and numerous other stress signals. This increase in p53 results in the activation and transcription of various genes that play important roles in cell cycle arrest, senescence, apoptosis, metabolism, and differentiation [23]. P53 induces cell cycle arrest and multiple DNA repair mechanisms. The cell-cycle arrest triggered by DNA damage is reversible following the repair of DNA and p53 downregulation [22]. The functional inactivation of TP53 is quite common in human tumors, such as cervical cancer tumors. It is estimated that TP53 mutations occur in over 60% of such tumors, while in other types of tumors, functional inactivation of p53 may result from the activity of viral proteins as well as changes in other interacting pathways [24,25]. The loss of p53 function promotes cancer cell survival, tumor growth, invasion, metastases, and chemotherapy resistance [26]. Additionally, molecular interactions between mutant p53 and the tumor microenvironment have been described in the literature. These interactions result in the remodeling of the cancer microenvironment through the modulation of extracellular matrix components, the secretion of inflammatory proteins, and the interaction of cancer and stromal cells [24,27]. Banister et al. have analyzed HPV DNA and E6/E7 oncogene expression in the tumor tissues of patients with cervical cancer. All of the examined cervical cancer samples contained HPV DNA, and the majority of these expressed E6/E7 (E6/E7 mRNA) (HPV-active tumors). However, in 8% of cases the tumor cells taken from HPV-DNA-positive cervical cancer tumors did not express HPV-transcripts (HPV E6 and E7 oncogenes) (HPV-inactive tumors) [28]. HPV-inactive tumors occurred in older women and were associated with poorer rates of patient survival. The cancer driver genes (TP53, PTEN, and ARIDs) were more likely to be mutated in HPV-inactive tumors. It was suggested that among the female population, a small percentage of women may acquire somatic mutations in cancer-driven genes as a way to activate HPV-inactive tumors [28]. Abboodi et al. have observed that the complete loss of p53 function may contribute to the process by which HPV active lesions become inactive. In hr-HPV-positive tumors, p53 degradation induced by E6 eliminates p53, but this type of degradation is reversible. By contrast, complete loss of p53 is achieved when gene deletion alters the cell to the extent that it acquires an irreversible p53 null status and E6 function becomes redundant [25]. Such a change may contribute to the process whereby normal HPV-infected cells are transformed into cancerous cells. Furthermore, the authors stressed the possible prophylactic role of HPV vaccines in this transformation process [25]. 

### 1.3. HPV Infection

Human papillomavirus is a tissue-specific intraepithelial pathogen that can infect and replicate only within a fully differentiated squamous epithelium. The HPV replication cycle involves the infection of basal keratinocytes in an abrased epithelium [29,30]. In the dividing keratinocytes, the expression of viral genome copy numbers is tightly controlled, remaining at a low level (50–100 copy numbers). When the keratinocytes stop dividing and become mature cells, the expression increases, which induces the activation of all viral gene expression, including L1 and L2. Thousands of viral genome copy numbers are encapsidated and exit cells as infectious viral particles [31,32]. The viral replication process depends entirely on the keratinocyte cell cycle as the cellular DNA-polymerases and replication factors are produced only in mitotically active cells. HPV encodes the proteins responsible for the initiation of cellular DNA synthesis in noncycling cells, inhibits apoptosis, and delays the differentiation program [33]. In hr-HPV infections that are in an early stage of malignancy, the copy numbers for genes E6 and E7 are strictly controlled at the stage of cell division. When a keratinocyte stops dividing, the control of E6 and E7 gene expression is lost. The integration of HPV DNA into the host genome is facilitated by deregulated E6 and E7 expression. When integration disrupts the E1 and E2 regions, persistently high levels of E6 and E7 expression occur, resulting in an increased accumulation of genetic errors in the host genome [31,32]. This allows viral DNA replication in noncycling cells, leading to the deregulation of cell growth control [32]. All viral genes are expressed, and thousands of viral particles are released into the upper layers of the epithelium. The time from infection to the generation of infectious particles is approximately 3 weeks [32]. As the virus does not cause viremia, host cellular death, or local inflammation, the HPV infection remains undetected by the host. Furthermore, the virus does not cross the basement membrane where the more pronounced immune cell infiltration would stimulate an immune response. The viral particles are present in the upper layer of the epithelium where immune cell infiltration is uncommon. All the mechanisms mentioned restrict immune recognition of the HPV infection. Additionally, poor anti-HPV humoral immunity is observed in patients infected by HPV [32,33].

### 1.4. HPV-Negative Cervical Cancers

According to many studies, HPV infection contribution in the development and progression of invasive cervical cancer is quite common; however, 10–15% of invasive cervical cancer tissue samples test HPV-negative [34,35,36]. The differences in HPV-positivity were observed by using tumor histology. The highest rates of HPV-positivity were observed in the tissues derived from adenocarcinoma in situ (93.9%), adenosquamous carcinoma (85.6%), and usual-type adenocarcinoma (90.4%). By contrast, lower rates of HPV were observed in tissues representing rare adenocarcinoma subtypes [clear cell (27.6%), serous (30.4%), endometrioid (12.9%), and gastric-type (0%)] [37]. Negative HPV testing may result from the presence of false-negative results, low viral load, or the presence of other, not detected types [38]. From a cohort of 209 women suffering from cervical cancer, Kaliff et al. examined 37 tumor tissue samples that had previously tested negative or invalid. The authors then reinvestigated the tissue samples using repeated genotyping. After this reinvestigation, the proportion of HPV-negative cases could be reduced to one half of the samples tested [35]. The most common reasons for the HPV-negativity of the samples were linked to methodological aspects, such as tumor sample quality, storage time, and the loss of the viral L1 gene after viral integration [35]. Confirmed HPV-negativity was significantly associated with a worse prognosis, a more advanced patient age, a longer storage time, and an adenocarcinoma histology [35]. Bogani et al. observed a cohort of 2966 women in whom high-grade cervical dysplasia (HSIL/CIN2/CIN3) had been diagnosed. Patients with high-risk HPV-negative cervical dysplasia accounted for about 15% of the population studied. The prevalence of high-risk HPV-negative disease in those patients diagnosed with CIN2 was similar to the prevalence in those diagnosed with CIN3. Moreover, the high-risk HPV-negative patients were more likely to experience better outcomes than high-risk HPV-positive patients [36]. Independent of the type of treatment method applied (laser conization or loop electrosurgical excision procedure), the persistence of HPV infection was found to be the sole factor determining prognosis [38]. Although the persistence of HPV infection is responsible for the prognosis and recurrence of the disease, the literature has suggested that vaccination against HPV in women who undergo conization may reduce the risk of recurrent cervical dysplasia [39]. Bogani et al. analyzed a group of 116 women who underwent conization and vaccination as well as a group of 1798 women who underwent conization alone. The five-year recurrence rate was 1.7% in the group who were treated with both conization and vaccination, and 5.7% in the group treated with conization alone. Although vaccination did not influence persistent lesions, it did reduce the risk of recurrent disease. The study concluded that patients undergoing conization and vaccination remain at a slightly lower risk of recurrence than patients who have not been vaccinated [40].

## 2. The Role of Keratinocytes and Chronic Inflammation

HPV infects mucosal epithelia, targeting the dividing cells in the basal epithelial layer. In epithelial tissue stem cells, the HPV genome is deposited in the nucleus and begins to express early viral proteins. When infected basal epithelial cells divide, the viral genome copies are replicated into daughter cells. The continued presence of the viral genome over a period of several years in actively dividing epithelial cells results in persistent infection [8]. The keratinocytes of the cervical epithelium play an important role in HPV infection. On the one hand, keratinocytes allow, both in vivo and in vitro, the replication of many viruses, including human papillomaviruses (HPVs), by amplifying the viral load and facilitating viral spread [41]. On the other hand, keratinocytes behave like immune cells that can initiate both an innate and an adaptive antiviral immune response to fight viral infection [42]. Because of the expression of a wide range of Pattern-Recognition Receptors (PRRs), which transduce signals and initiate the transcription and secretion of proinflammatory cytokines, keratinocytes are sometimes recognized to be “immune sentinels” [41,43]. PRRs include cell-surface, endosomal, and transmembranal Toll-Like Receptors (TLRs). The activation of these PRRs triggers signaling pathways, leading to the production of interferons (IFNs), proinflammatory chemokines, cytokines, and antimicrobial peptides, all of which represent key players in the innate immune response [41]. Secreted proinflammatory mediators are involved in recruiting monocytes, macrophages, polymorphonuclear neutrophils, and dendritic cells to the site of viral infection. The HPV infection interferes with the immunomodulating function of keratinocytes, which allows it subsequently to evade the immune response. The HPV infection induces the keratinocytes to secrete proinflammatory cytokines. High-risk HPV E6 and E7 interact with interferon regulatory transcription factor (IRF) and inhibit the cells’ transcriptional activity [44]. They also inhibit the activation of TNF-receptor-associated factor 3 (TRAF-3) by upregulating Ubiquitin C-terminal hydrolase L1 (UCHL1) [45]. Furthermore, HPV E7 inhibits the functional effects of INF-alpha by targeting p48, resulting in the loss of IFN alpha-mediated signal transduction [46]. 

Interleukin-6 (IL-6) is a proinflammatory cytokine that inhibits the apoptosis of cells during inflammation through the activation of JAK-STAT signaling pathways after binding to the IL-6 receptor (IL-6R) [47,48]. IL-6 has been reported to be upregulated in patients with many types of cancers, including cervical cancer, and is considered to play an important role during tumorigenesis. In women with cervical cancer, high levels of IL-6 expression in the cancer microenvironment promote angiogenesis and the development of cancer [49]. IL-6 has also been shown to be a target of E6 and E7 oncoproteins, which upregulate IL-6 expression in keratinocytes. IL-6 has been found to be overexpressed in CIN1 tissue samples in comparison to control cervical samples without lesions. Additionally, IL-6R is overexpressed in tumoral proliferative keratinocytes in cervical cancer tissues. Therefore, viral oncogenes E6 and E7 would seem to stimulate chronic inflammation in the cancer microenvironment through IL-6 and IL-6R overexpression [50]. Additionally, viral oncogenes E5, E6, and E7 increase cyclooxygenase- (COX-) 2 expressions, which can lead to the release of large numbers of prostaglandins that stimulate cell proliferation and angiogenesis while inhibiting apoptosis. Accordingly, these oncogenes participate in chronic inflammation and carcinogenesis [51].

## 3. Immune Cells

The cervical cancer tumor is infiltrated by tumor-infiltrating lymphocytes (TIL), meaning CD3+, CD4+, and CD8+T cells, and CD3+CD4+ (double positive) T cells are predominant in the tumor center [52]. An increased infiltration of CD3+ T cells, CD4+ T cells, and CD8+ T cells in the tumor center has been shown to correlate with poor prognosis [52]. The tumor progression was related to an increase in the number of TIL (CD3+, CD4+, and CD8+ T cells) [52]. The hr-HPV was observed to affect the activity of CD8+ T cells and memory CD8+T cells. It was suggested that both the density and distribution of immune T cells depend on the malignant potential of hr-HPV lesions [53,54]. Zou et al. observed that the T cells identified in the invasive margins and tumor center had an antitumor potential, while over time the tumor cells were able to escape the CD4+T cells immune surveillance by modifying their own surface antigens [52]. 

Within the tumor microenvironment at virus-infected sites, a suppressive profile is created through hypoxia and oxidative stress. This process is regulated by various cells and mediators. It has been found that HPV may interfere with the Th1/Th2 polarization of the immune response. In early lesions induced by HPV-infection, a Th2-like immune response with dominating cytokines, such as IL-6 and IL-10, has been observed. In high-grade epithelial lesions, TNF-alpha levels are upregulated, and both Th1- and Th2-response cytokines are increased. The change in the profile of the immune response from Th1 to Th2 during the disease’s early stages may influence its persistence and progression [55,56,57,58]. 

Regulatory and suppressive immune cells can be recruited to the virus-infected sites and cancer microenvironments through various factors, including cytokines IL-6 and IL-10, as well as growth factors like VEGF and PGE2. Tumor-associated macrophages present in the tumor microenvironment are strongly associated with viral infection [59]. Increased infiltration of CD68+ and CD163+ macrophages was associated with hr-HPV infection and positively correlated with cervical carcinogenesis. Furthermore, the existence of a large number of CD163+ macrophages was significantly linked to an advanced FIGO stage and to lymph node metastasis [60]. Myeloid-derived suppressor cells (MDSCs) are a heterogeneous population of immature myeloid cells that can strongly inhibit the anti-tumor activities of T and NK cells and stimulate regulatory T cells (Treg). MDSCs are generated under chronic inflammatory conditions by the release of soluble mediators into the tumor microenvironment or by extracellular vesicles. These MDSCs include IL-6, IL-10, IL-1beta, granulocyte-macrophage colony-stimulating factor (GM-CSF), granulocyte colony-stimulating factor (G-CSF), macrophage-colony stimulating factor (M-CSF), chemokines (chemokine (C-C motif) ligand 2 (CCL)2, CCL5, CCL26, and chemokine (C-X-C motif) ligand 8 (CXCL)8), CXL12, and prostaglandin E2 (PGE2). The secretion of interleukin (IL-10) and interferon (IFN)-γ mediates the generation of M2 macrophages and Treg cells [61].

## 4. HLA-G as an Antigen Participating in the Creation of the Suppressive Cancer Microenvironment in Cervical Cancer Patients

The progression of cervical cancer is determined by a complex interaction between the host’s immune system and the HPV infection. Only a small percentage of HPV infections develop into high-grade squamous intraepithelial lesions (HSIL)- CIN3 (10% of HPV infections) and cervical cancer (less than 1% of HPV infections). Many studies have analyzed HLA-G expression in cervical cancer patients. Human leukocyte antigen-G (HLA-G) is one of the nonclassical HLA-class Ib molecules associated with various mechanisms of immune suppression. For the first time, HLA-G has been identified in trophoblast cells as a ligand for suppressive receptors present on uterine NK cells and participating in the process of immune tolerance during pregnancy [62]. Malignant neoplasms use various mechanisms to escape from the host’s immune control; the wide range of HLA-G activity suggests that it plays an important tolerogenic role in this process [63]. Selective HLA-G expression allows not only viruses, but also cancer cells, to escape from immune host control and elimination. Specifically, HLA-G inhibits the activity of cytotoxic T lymphocytes and NK cells and, consequently, their proliferation and antigen-presenting cell maturation. At the same time, HLA-G induces the suppressive functions of T-regulatory lymphocytes, activated T CD8+ lymphocytes, and NK cell apoptosis [62,63]. Furthermore, the HLA-G molecule stimulates CD4+ lymphocytes to produce cytokines (IL-3, IL-4, and IL-10) by suppressing the function of cytotoxic T lymphocytes [64,65]. As a result of this cytokine production, the Th1/Th2 balance shifts to a Th2-dependent response [65]. Soluble forms of the HLA-G molecule are also characterized by immunosuppressive activity; they inhibit NK-dependent cytotoxicity and IFN-gamma secretion [66]. The HLA-G5 molecule stimulates the differentiation of immature T lymphocytes into Treg cells with lower levels of both CD4 and CD8 expression and decreased reactivity. The soluble forms of HLA-G induce CD95 expression as well as the apoptosis of T lymphocytes and NK cells in a Fas/FasL pathway. The interaction between HLA-G and dendritic cells disrupts the latter’s maturation and migration [66,67,68]. Table 1 presents the tolerogenic functions of the HLA-G molecule and its interactions with immune system cells [62,63,66,67,68]. Dong et al. have analyzed HLA-G expression in patients with and without HPV infection, observing that the levels of HLA-G expression increased progressively from those found in the CIN1 tissues to those in the CIN2/3 tissues, and were at their highest levels in the squamous cell carcinoma tissues. Furthermore, HLA-G expression levels were statistically significantly higher in the tissues of the CIN patients with HPV-16/-18 infection compared to the CIN patients without HPV infection. Hence, the authors concluded that HLA-G expression is involved in HPV infection and tumor progression [69]. Yoon et al. have looked at the expression levels of mRNA HLA-G and IL-10 in patients with cervical carcinogenesis. The levels of both HLA-G and IL-10 expression in the tissue samples of cervical cancer patients were statistically significantly higher than in the control tissue samples. Similarly, the levels of HLA-G and IL-10 proteins were higher in cancer tissue samples than in normal control samples, but the difference was not statistically significant. Additionally, the HLA-G mRNA expression levels were higher in the tissues of patients in the early stages of the disease compared to those with advanced disease. Accordingly, it was concluded that HLA-G expression may participate in the process of carcinogenesis and cervical cancer progression [70]. Zheng et al. analyzed HLA-G expression levels in normal tissues, CIN1-CIN3 tissues, and cervical cancer tissues. No HLA-G expression was detected in the normal tissue samples. HLA-G expression levels were statistically significantly higher in the CIN and cervical squamous cell carcinoma tissue samples than in normal tissue samples. Moreover, a statistically significantly higher HLA-G expression was correlated with tumor size, involvement of the parametrium, and the presence of lymph node metastases. The soluble form of HLA-G protein was also analyzed in the blood sera of patients with CIN2, CIN3, and cervical squamous cell carcinoma. A statistically significantly higher level of sHLA-G was found in the tissues of patients with CIN2, CIN3, and cervical squamous cell carcinoma when compared with controls; however, no such differences were observed in the tissues of healthy individuals compared to patients with CIN1. These findings suggest that evaluating the presence of sHLA-G in the blood serum could prove useful for early cervical cancer detection [71]. Li et al. have confirmed a positive correlation between sHLA-G expression and carcinogenesis in patients with cervical cancer [72]. HLA-G, IL-10, and HLA-Ia expression levels were analyzed in the tissues of patients with CIN and with cervical squamous cell carcinoma. Analysis showed that IL-10 secretion may stimulate the local immune suppressive microenvironment in patients with cervical cancer by upregulating HLA-G expression and downregulating HLA-Ia expression within the microenvironment. This process could result in a decreased susceptibility to NK- and cytotoxic-lymphocyte-mediated immunity [73].

Finally, HLA-G polymorphism was studied, revealing that in women in Brazil, CIN3 polymorphism of HLA-G was linked to HPV infection and CIN3. The studies concluded that the presence of HLA-G polymorphism created a profile compatible with a predisposition to cervical cancer development [74]. Researchers have also examined the relationship between HLA-G polymorphism and susceptibility to HPV infection and its chronic course. Studies have confirmed that HLA-G may participate in the process of controlling HPV infection risk, and the presence of various HLA-G polymorphisms could potentially influence the development of cervical lesions [75]. Upon further research, the authors determined that HLA-G may participate in the progression from preinvasive lesions to invasive squamous cell carcinoma of the cervix. They concluded that HLA-G polymorphism is a strong and independent risk factor for cervical cancer development. This finding proves that HLA-G has a role in the creation of the suppressive cancer microenvironment that allows the tumor to escape from the host’s immune control [65]. Metcalfe et al. have shown that HLA-G polymorphism may play an important part in the natural history of the HPV infection, likely during the early stage of host immune recognition [75,76,77]. Lastly, Bortolotti et al. have confirmed that HLA-G polymorphism (HLA-G 3’ untranslated region polymorphisms (14 bp ins/del, +3142 C > G) promotes hr-HPV infection, with del/C haplotype associated with invasive cervical cancer development [78].

In summary, an infection of HPV is an unquestionable causative factor for cervical cancer, however, it is not well understood how the virus persists for decades before the development of cancer; it seems that the virus needs mechanisms to escape host immunity. The HLA-G protein expression was upregulated in tissues from the HPV-positive cervical carcinoma, as compared to the HPV negative normal cervix [69,71,72,73,79]. The increased expression of HLA-G seems to intensify the persistence of HPV infection [79]. On the one hand, HLA-G expression might serve as a marker of malignancy in cervical cancer, both the HLA-G membrane form expressed by cervical cancer cells and the HLA-G soluble form identified in blood serum. On the other hand, it might also become a therapeutic target for strategies based on anti-HLA-G antibodies to eliminate HPV infection [79].

## 5. Potential Application of Post-Infection Local Microenvironment Biomarkers in New Diagnostic and Therapeutic Strategies

The local, post-infection, immunosuppressive, virus-supporting microenvironment seems to have become a novel yet important diagnostic and therapeutic target. This diagnostic and therapeutic strategy focuses on the identification of post-infection microenvironment biomarkers and the on inhibition of post-infection microenvironment formation. 

At present, the research is performed to identify biomarkers reflecting the formation of the local post-infection microenvironment. This includes the identification of the immune cells and their activity and the presence of the cytokines, and chemokines, participating in suppressing the immune responses (IL-6, IL-10, IFN-alpha, IFN-beta, IFN-kappa, IFN-gamma, CXCL4, CCL20) [41,42,43,44,45,46,47,48,49,50], as well as the factors determining oxidative stress and inflammation (PGE2, ROS, iNOS) [51], the factors of extracellular matrix remodeling (PDGF, FGF, uPA, MMPs, TIMP-2) [47], and the factors of angiogenic switch (VEGF, HIF-1alpha, IL-8) [52,53,54,55,56,57,58,59,60,61,80].

The targeting of the local immunosuppression encompasses the reactivation of antiviral immune response and the reversal of the local immunosuppression. Immune checkpoint inhibitors (ICIs) are one of the therapeutic targets, having been used in the treatment of melanoma, Hodgkin’s lymphoma, and bladder cancer. PD-1 (programmed cell death protein 1) and PD-L1 (programmed death-ligand 1) inhibitors have been approved by the U.S. Food and Drug Administration for clinical use [81]. PD-1 and PD-L1 interrupt the interaction between effector T cells, cancer cells, and dendritic cells, inhibiting T cell growth [82]. PD-1 and PD-L1 inhibitors like Pembrolizumab, Nivolumab, Atezolizumab, and Durvalumab and the cytotoxic T-lymphocyte antigen-4 (CTLA-4) inhibitors, like Ipilimumab and Tremelimumab, seem to be promising therapeutic targets in reversing the immune suppression [83,84].

IFNγ and TNFα suppress the proliferation of keratinocytes; IFNγ induces growth arrest and differentiation, and subsequently induces caspase-mediated apoptosis [85,86]. Ma et al. have demonstrated that hrHPV-infected keratinocytes become resistant to the IFNγ- and TNFα-induced necroptosis and cell growth arrest, which was related to the downregulation of RIPK3 (receptor-interacting protein kinase 3) expression. Smola has concluded that the RIPK3 expression might be a potential biomarker of prediction for response to dsRNA-based immunotherapies such as PolyIC-derivatives [87].

In conclusion, the development of novel biomarkers and therapies in cervical cancer has to consider the deep influence on the microenvironment remodeling related to HPV infection.

## Figures and Tables

**Table 1 diagnostics-12-01906-t001:** Tolerogenic functions of HLA-G—an interaction of HLA-G with immune system cells [62,63].

Immune Cells	Receptors	Inhibition	Activation
Dendritic cells	ILT2 (inhibitory receptor Ig-like transcript 2) ILT4 (inhibitory receptor Ig-like transcript 4)	Dendritic cells’ maturation	Tolerogenic dendritic cells
NK cells	ILT2 KIR2DL4 (Killer Cell Immunoglobulin Like Receptor, Two Ig Domains and Long Cytoplasmic Tail 4)	Cytotoxicity IFN-gamma secretion MICA/NKG2D activation chemotaxis	Th2 cytokines HLA-E expression VEGF production FasL mediated apoptosis
T lymphocytes	ILT2 KIR2DL4	Proliferation Cytotoxicity INF-gamma secretion of gamma delta T cells Cytolysis Chemotaxis	Th2-type cytokine Treg FasL mediated apoptosis
B lymphocytes	ILT2 ILT4	Proliferation Ig and cytokines secretion chemotaxis differentiation	FasL mediated apoptosis Th2-type cytokines

## Data Availability

Not applicable here.

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
