# Peer review of "The Creation of the Suppressive Cancer Microenvironment in Patients with HPV-Positive Cervical Cancer"

_diagnostics, 2022, doi:10.3390/diagnostics12081906_

Round 1
Reviewer 1 Report
In the manuscript entitled “The creation of the suppressive cancer microenvironment in patients with HPV-positive cervical cancer”, the authors reviewed the underlying mechanism of chronic HPV infection and HPV-positive cervical cancer. The authors also focused on the remodeling of the microenvironment induced by HPV infection through the epithelium, chronic inflammation, and cytokines. Additionally, the authors emphasized the role of HLA–G as an antigen participating in the creation of the suppressive cancer microenvironment in cervical cancer.
The review is well received and there are adequate references to support the conclusion. There are only minor spell checks required.
Author Response
According to the Reviewer suggestion the language correction of the manuscript was provided by Dr Christine Maisto, the native speaker and professional medical English editor (California University).
Reviewer 2 Report
Thank you for allowing me to review this paper.
The paper reports interesting data.
The discussion should be implemented by adding data on the role of HPV-negative disease. Some cervical cancer and cervical dysphasia are HPV-negative. Low viral load, other HPV types not detected by conventional methods, false negative results, and no HPV-related disease are the main cause. Please see this paper (PMID: 33514481).
More interestingly is the association between HPV persistence and the risk of developing high-grade cervical dysplasia and cervical cancer. These papers are referring to this (PMID: 3545532, PMID: 33271963, PMID: 32893030). The association between host immune response and HPV persistence is strong. but few data are available.
Author Response
According to the Reviewer suggestion an additional paragraph concerning HPV-negative cervical cancers was implemented to the study with proper literature (paragraph 1.4).